# More Than Eleven Thousand Words: Towards Using Language Models for Robotic Sorting of Unseen Objects into Arbitrary Categories

**Sigmund Hennum Høeg and Lars Tingelstad**
Department of Mechanical and Industrial Engineering
Norwegian University of Science and Technology
{sigmund.hoeg, lars.tingelstad}@ntnu.no

**Abstract:**

We consider the task of automatically sorting previously unseen objects into arbitrary categories. We aim to sort into general, high-level categories in contrast to traditional methods that sort on visually discernible features or by other sensor measurements. This paper explores a method where we divide the categorization into two sub-tasks: object detection and categorization. In a set of experiments, it is shown that splitting the categorization task into a two-stage process removes highly important information for robust categorization and performs less robustly than an open vocabulary object detector. We hope these results are helpful for exploring the limits of Language Models for robotic tasks.

**Keywords:** Language Models, Detection, Learning, Sorting

## 1 Introduction

We recognize the ubiquity of sorting tasks. From industrial settings, second-hand stores, and household services, sorting objects provides value. So far, however, many sorting tasks have been restricted to human execution. Considering this, we consider the question: *Can we devise a robotic system that can sort objects efficiently and in a highly general manner?*

A usual approach to sorting systems is training on a fixed set of categories, such as color, material, or other features measurable with sensors or cameras. Compared to humans, these methods suffer from two significant distinctions that hinder their flexibility: (i) they need to be retrained when presented with new categories, and (ii) the categories must be discernible from sensor input. These characteristics limit the use of robotics in cases where categories might change often and are provided as high-level expressions. Consider a service robot in a specific household where the pots and pans are stored in one drawer and children's toys in another. Reprogramming the sorting system to recognize these two specific categories is limiting. Another application is second-hand stores, such as Goodwill, which might have specific categories they sort after, determined by factors such as target groups, seasons, or campaigns. These categories might change rapidly, and to compete with humans, the sorting systems should allow rapid switching between categories. From an environmental perspective, improving the throughput of the reuse industry will positively impact the circular economy and lower the demand for production.

An apparent challenge is that the categories are previously unseen, separating this from traditional classification in computer vision. We must capture the relation between an arbitrary object and an arbitrary category. A challenging aspect of this task is the wide range of objects and categories we can encounter. For many categories, modeling the relationship between the sensor observation and category membership relies on a complex semantic understanding. Consider the class of children's toys, a category containing objects of all shapes, weights, colors, and sizes.

6th Conference on Robot Learning (CoRL 2022), Auckland, New Zealand.

Several works have recently shown that combining Language Models (LMs) with robotics systems allows for understanding natural language and complex reasoning for long-horizon planning. [1, 2] An emerging challenge is ensuring that the proposed actions are executable in practice. Given the problem of flexible and high-level sorting and the advancements of LMs for robotics, we want to answer the pertinent question: *Can we incorporate the semantic understanding of LMs into a robotic system to achieve efficient yet flexible sorting behavior?*

The sorting task can be separated into the following tasks: (i) Detect a single object in the scene and determine its category and subsequently (ii) determine pick-and-place positions, and lastly (iii) execute the pick-and-place action with the robot. We recognize that detection and categorization is the primary limiting factor for allowing a more extensive set of classes and objects. Responding to this challenge, we explore an approach for categorization by separating the task into two steps. First, perform classification, namely assigning the membership to a specific class, and categorization, assigning this class to a higher-level category. Figure 2 shows a schematic view of this approach.

We conduct experiments on images to measure the method's performance compared to directly using open-vocabulary classifiers. We present and discuss failure modes and promising tactics to improve the method. In general, the results show that it is beneficial to predict the category directly from the image, pointing to the fact that visual appearance provides essential cues for determining the category of the object.

Section 2 will provide an overview of the related work, Section 3 will describe the approach of our method, Section 4 will present experimental results, and Section 5 will conclude and point to further research.

## 2   Related work

**Robotic sorting** Several systems perform sorting of objects into predefined categories; however, to the best of our knowledge, no previous work has considered the problem of open vocabulary robotic object sorting. Several works have investigated the sorting of objects based on specific properties such as color, shape, and material. [3, 4, 5] An application for these methods is waste sorting, where Lukka et al. [6] performs sorting by material properties. Similarly, Kujala et al. [7] sorts objects by color off a conveyor belt.

**Automatic sorting** Guérin et al. [8] explore the problem of sorting objects without specified categories. They use a convolutional neural network in combination with a clustering algorithm to group objects into a given set of bins. Their method is, however, restricted to sorting objects of similar appearance. The sorting system is also unaware of the semantic meaning of the different classes, and the system does not allow the operator to specify categories.

**Language models for robotics** Zeng et al. [2] show that combining LMs with Visual Language Models (VLMs), allowing them to communicate through language, results in an overall system with a high understanding of the scene. Specifically, they use VLMs to inform the system what objects are in the scene, and a prompt format guides the LM to output code-like responses, e.g., pick-and-place actions. Ahn et al. [1] compares the predictions from the LM to affordance functions, predicting the most relevant action given the robot's surroundings and the given task description.

**Object retrieval** Nguyen et al. [9] focus on the problem of retrieving the most relevant object given a command containing a verb, imbuing the robot with semantic understanding. Their approach is limited to retrieval given an action that can be done with the object, whereas our approach considers a free space of categories.

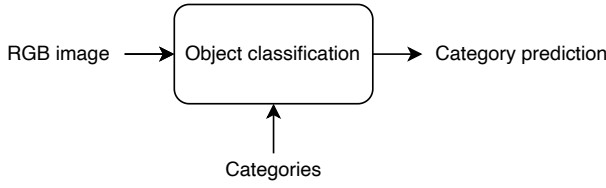

Figure 1: Schematic for the approach of direct categorization.

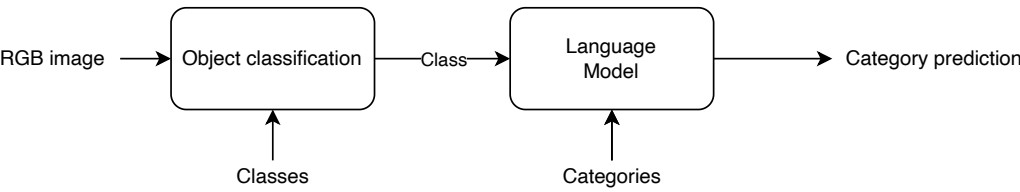

Figure 2: Schematic for the approach of separating classification and categorization.

# 3 Proposed method

We will discuss two approaches to performing object categorization:

1. Categorizing directly using an open vocabulary object detector.
2. First, detect objects using a large vocabulary and then categorize them using a language model.

These will be described in detail in Section 3.1 and Section 3.2.

## 3.1 Categorization directly

As shown in Figure 1, we aim to categorize the objects directly, meaning we find the associated category without classifying the objects first. This aligns with the task of an open-language object detector, which can detect objects using an arbitrary vocabulary.

We hypothesize that this method will work well for categories within its training distribution. This includes categories on the same level of abstraction. Perhaps classes such as "cat," "poodle," and "bicycle" are more straightforward to recognize than "travel-related," "furniture," and "kids toys"?

## 3.2 Separating detection and categorization

An overview of this approach is shown in Figure 2. In the spirit of Ahn et al. [1], we incorporate an LM in the task of assigning the object to a category. We achieve this by splitting the task into classification and categorization. In short, we use an object classification model to obtain $P(\text{class}|\text{image})$, where "class" is an class in the vocabulary of the classification model (e.g. "apple"). Subsequently, we query an LM to obtain $P(\text{category}|\text{class})$, where "category" is one of the given categories (e.g. "food").

We can express the original task of estimating the category $C$ from an image $I$ as finding:

$$C^* = \arg\max_C P(C|I).$$

Incorporating an LM, we reformulate the categorization problem to depend purely on the object's class. This implies assigning two attributes to the object in the image $I$, namely the class of the object $O$ and its corresponding category $C$. We are then interested in maximizing the following:

$$(C^*, O^*) = \arg\max_{C,O} P(C \cap O|I).$$

We write the above expression as the two conditional probabilities:

$$(C^*, O^*) = \arg\max_{C,O} P(C|O \cap I)P(O|I).$$

We now make two assumptions to allow the separation of concerns. First, we assume that $P(O^*|I) = 1$, meaning that we have a perfect object classifier predicting only one relevant class. The maximization is then only over the categories. Secondly, we assume that information about the object's class is sufficient to estimate the category, meaning that the category $C$ is conditionally independent of the image $I$. We simplify the expression to:

$$\hat{C}^* = \arg\max_C P(C|O). \tag{1}$$

Here, classification means detecting the objects in the scene and classifying them into a set of classes. The classes can be fixed, and the classifier can be trained specifically for them. Therefore, this task can be performed by an object detector for separate objects or an image classifier for the whole image. The instance masks from the former help predict grasps downstream. For categorization, we can use an LM to get the probability that an object class is assigned to a given class by careful prompt engineering. The details are provided in Section 4.

We hypothesize that this two-stage approach is beneficial when it is challenging to determine the category given the detected object class. This might be the case when reasoning is required to determine the category. For example, it might be easier for an LM to capture that and "apple" should be in the "food" category than it is for an object detector to detect something as "food" in the image directly.

## 4   Experiments

Here, we present experimental results from each of the presented methods in Section 3 for categorizing objects. We benchmark the methods for classifying images of singular objects into a set of given categories.

We use an open-vocabulary object detector and an open-vocabulary image classifier for the direct categorization approach. Specifically, we choose ViLD [10] and CLIP [11], both representing state of the art on several datasets. Pre-trained models are publicly available for both algorithms.

For the two-step approach, we conduct experiments with ViLD and CLIP using a fixed vocabulary for object detection. GPT-3 [12] is used as the LM predicting the category given the detected class. As we want the object detector to be as specific as possible, we use the class labels from the Tencent-ML Images Database [13] as this vocabulary. It combines the categories from both ImageNet11k [14] and Open Images [15], resulting in $11\,166$ categories. We do not evaluate a fixed vocabulary object detector, nor do any fine-tuning of the object detector on the Tencent-ML dataset. This might help improve this approach.

We use GPT-3 with the following prompt:

```
prompt = "Classify each of the following objects as either "
prompt += ", ".join(category_list[:-1])
prompt += " or " + category_list[-1] + ". "
prompt += "Object: " + detected_object + " Label:"
```

where `detected_object` is the detected object class, and `category_list` is the each of the provided categories. We make a prompt for each category, where the relevant category is appended to the prompt. The resulting category is associated with the prompt with the highest probability from the language model.

### 4.1   Images in the wild

To indicate the robustness of the two methods, we sample 23 images from the web and measure the categorization accuracy for each method.

Table 1: The different sets of categories used in the experiments.

| Narrow categories | Broad categories | YBC categories |
|---|---|---|
| newer books | eating related | food items |
| LP records | entertainment | kitchen items |
| CDs and DVDs | decorative | tools |
| small interior items | | |
| crockery | | |
| mugs and glasses | | |
| serving bowls | | |
| cutlery | | |
| vases | | |
| ornaments | | |
| candlesticks | | |
| tablecloths | | |
| decorative pillows | | |
| wool blankets | | |
| pictures and paintings | | |
| kitchen utensils | | |
| small furniture | | |
| toys and games | | |
| working electrical items | | |
| working kitchen equipment | | |

Table 2: The accuracy of all methods for the the different sets of categories

| Method | Narrow categories | Broad categories | YCB images |
|---|---|---|---|
| ViLD | 45.5% | 40.9% | 63.8% |
| CLIP | **86.3%** | 57.1% | **91.5%** |
| ViLD+GPT-3 | 4.55% | 36.4% | 55.3% |
| CLIP+GPT-3 | 45.5% | **76.2%** | 63.8% |

Motivated by the second-hand industry, we sort the images into categories that might be used when receiving second-hand goods. These are listed in the first column of Table 1. We label each image by the most suitable category.

## 4.2 Results

The accuracy for categorization into the narrow categories for each method is listed in the first column of Table 2. For both CLIP and ViLD, the unified one-step approach achieves a higher result, an order of magnitude in the case of ViLD, at 45% compared to 4.5%. Figure 3 and Figure 4 show the result on a sample of images using the one-step and the two-step approach, respectively. Here too, we see that the direct approach yields higher accuracy.

## 4.3 Categorization into broader categories.

Motivated by the hypothesis in Section 3.2 that the two-step approach might be more robust with more abstract categories, we test the methods on fewer and broader categories. These are listed in the second column of Table 1.

The second column of Table 2 lists the performance of all the approaches. In the case of ViLD, the direct method outperforms the two-stage approach. Note that, due to only having three categories, a random categorization method would yield an accuracy of 33%. For CLIP, however, we see that the two-step approach outperforms the one-step approach.

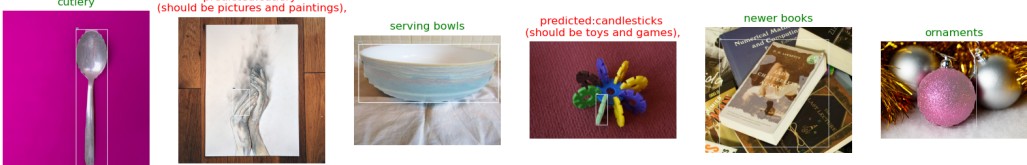

Figure 3: One-step categorization using ViLD

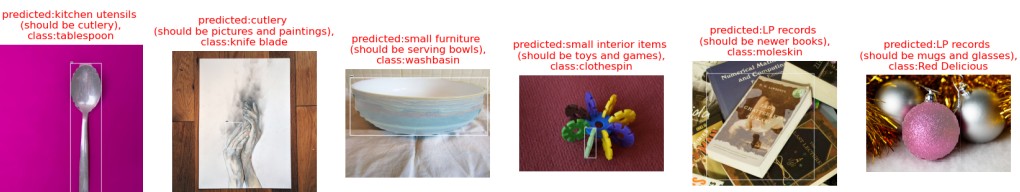

Figure 4: Categorization results using a two-step approach with ViLD.

## 4.4 Categorization of YCB-objects

We apply both approaches to images of objects in the YCB-dataset [16], which amounts to 47 images. The categories are listed in the third column of Table 1. The images of YCB objects typically include a single object against a neutral background, whereas the In-the-wild images have varying backgrounds and distracting objects.

The results are summarized by their accuracy in Table 2. The one-step approach shows superior performance for both CLIP and ViLD. The results of the two-step approach for a set of images are shown in Figure 5.

## 4.5 Classification and categorization evaluation

We evaluate the two parts of the two-step system in the following way. For classification, we note how frequently the main object in the image belongs to the predicted class. The results are summarized in Table 3. CLIP classifies substantially more robustly compared to ViLD, indicating the additional challenge of predicting instance masks and relevant bounding boxes.

For the classification by the LM, we count how many of the categories predicted by the LM include the given class. On average, over all experiments, GPT-3 achieves a categorization accuracy of 87.4%.

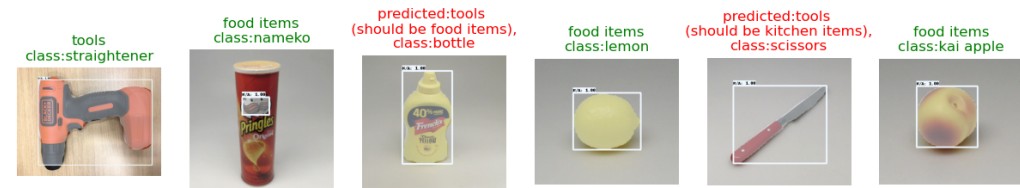

Figure 5: Categorization results using two-stage approach (ViLD+GPT-3) on the YCB objects.

Table 3: Classification accuracy for different VLMs, using the vocabulary of the Tencent-ML Database [13].

| Classifier | In-the-wild images | YCB images |
|---|---|---|
| CLIP | **45.5%** | **53.2%** |
| ViLD | 9.09% | 29.8% |

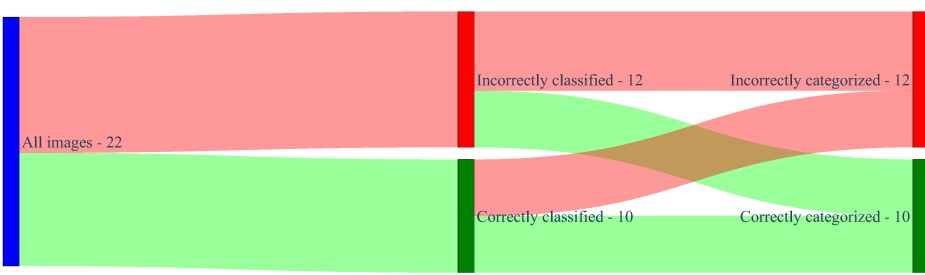

Figure 6: The flow of images through the 2-step approach with CLIP and GPT-3, with the In-the-wild images categorized into narrow categories.

## 4.6 Discussion

In the case of ViLD, a significant failure mode is inappropriate bounding boxes. It can capture an irrelevant part of the image or just part of the object of interest. When the bounding box is too small, the categorization is challenging due to the lack of context. For example, consider the children's toy in Figure 3, where only a small part of the object is included in the bounding box. This provides limited context for object categorization downstream. Among the tactics to improve this is fine-tuning hyperparameters for the bounding box prediction in ViLD.

While the one-step method either fails or succeeds in categorizing correctly, the two-stage can fail or succeed in several ways. This is illustrated in Figure 6, showing the flow of images through the two-step system. Here, there are two ways in which an image can end up miscategorized. The first is an initial misclassification, leading to a miscategorization by the LM. The other is when a correctly classified image is still incorrectly categorized.

Table 3 reveals that there is room for improvement in the classification step. Aside from a standard classification error where the detector chooses the wrong class, another cause for misclassification can be a lacking vocabulary. An example is the Christmas ball present in the rightmost image in Figure 4. The Tencent-ML Image Database does not include the class "Christmas ball", and the detector predicts a visually similar, but wrong, class. Improving the classifier or extending the vocabulary are tactics to reduce the number of classification errors.

The miscategorization of an already correctly classified image can be split into two cases. The first is an incorrect category prediction from the LM. We see empirically that this is relatively rare, as the LM correctly categorizes the class 87.4% of the time in our experiments. The other case is due to the predicted class lacking specificity, such as the class prediction for the mustard bottle in Figure 5. Here, the predicted class "bottle" is too vague for the LM to determine that this is a food-related bottle. A more specific classification, such as "mustard bottle" would improve the categorization accuracy. Again, simply improving the vocabulary is a promising approach to alleviate such errors.

On the other hand, the two-step approach can also succeed in two ways. As shown in Figure 6, some of the previously misclassified images are still correctly categorized by the LM. An example picture is shown in the rightmost subfigure in Figure 5, where a peach is incorrectly classified as an apple, which is also a member of the overall category "food".

Empirically, we see that the assumptions made to satisfy Equation 1 are not valid. Firstly, the classifier is imperfect, and propagating its uncertainty to the classification stage could be an exciting extension. Secondly, the conditional independence $P(\text{category}|\text{class} \cup \text{image}) \approx P(\text{category}|\text{class})$ is a faulty assumption, as a textual description of the object might lack sufficient information for downstream decisions. Specifically, the textual description might be too vague to provide the relevant information to categorize the object correctly.

## 5   Conclusion

In this paper, we compare two approaches for sorting objects in RGB images motivated by improving robotic sorting. We implement an approach where the task of determining the class of an object (e.g., "apple") is separated from categorizing the object into categories (e.g., "food"). In a set of experiments, we measure the accuracy of previously unseen images and categories. Additionally, we present insights into where the different methods fail and succeed. The results indicate that access to visual information is relevant to determining the category and that describing the object sufficiently by its class is restrictive. Therefore, a direct approach behaves more robustly in our experiments.

As directions for further work, improving direct, open-vocabulary categorization methods emerges as a promising direction. A study of the effect of prompt engineering could provide some interesting hints on how to improve VLMs in general. Additionally, in the case of second-hand stores, it is not only crucial to group objects into categories but also evaluate whether the object is in good condition and applicable for reselling. This new dimension is helpful to consider in further work to progress towards more effective second-hand sorting systems. Another point of further work is investigating the gain of using RGB-D images for classification, as such observations are typically available on modern robotic platforms. This might help increase the detection, classification, and categorization robustness.

**Acknowledgments**

The authors would like to thank the reviewers for their helpful feedback and interesting perspectives. This work was funded by the Norwegian Research Council under Project Number 237896, SFI Offshore Mechatronics.

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
