# OpenReview forum: "More Than Eleven Thousand Words: Towards Using Language Models for Robotic Sorting of Unseen Objects into Arbitrary Categories"
_robot-learning.org/CoRL/2022/Workshop/LangRob — LangRob 2022 Poster_

### Official Review · Reviewer_BFjT · 2022-11-11
**Interesting paper, suitable for the workshop**

**Rating:** 6
**Confidence:** 4

**Review:**

The paper addresses the problem of taking as input a set of images of various open-category objects, and sorting them into categories specified at test-time in natural language. Both the objects and categories are previously unseen. The work is motivated by the desire to build robotic sorting systems for practical applications, however it focuses specifically on the visual and language grounding challenge.

The paper compares two types of approaches to the problem: using one-step method that uses an open-vocabulary object detector (ViLD) to classify the objects, and a two-step method that splits the task into detection (also with ViLD) and categorization using a language model (GPT-3).

The main finding is that direct classification with ViLD works better, since using a pre-specified set of classes in the two-step method loses relevant information.

My suggestion to the authors would be to maybe try using a CLIP encoder directly on crops of objects. ViLD does this internally, but ViLD is also trying to solve the challenging problem of generating object masks of all sizes. This isn't needed just for categorization, but the masks can be helpful when interfacing with downstream robotics systems.

Pros:
- Addresses a relevant problem to robotics and language.
- Studies recent and relevant methods for the problem.
Cons:
- Isn't yet very extensive in studying the problem, other than trying two sensible approaches to the problem. It's ok for a workshop paper, but I would encourage deeper inquiry.
- Doesn't connect to any robotics systems, but that's probably ok.

---

### Official Review · Reviewer_yomD · 2022-11-13

**Rating:** 6
**Confidence:** 5

**Review:**

This is a neat paper that studies two possible ways to do open vocabulary category prediction: direct categorization with ViLD and categorization followed by LLM parsing. The finding is not particularly surprising the two-step process heavily relies on the initial classes to be valid and involves projecting the information into a fixed high dimensional space (rather than an open one). Some thoughts:

- I strongly encourage the authors to try this with vanilla CLIP, rather than ViLD. There are lots of works that found that ViLD performs worse than CLIP in the basic detection task (because of multiple objectives and the focus on bounding box pred etc.). For instance, see [LM-Nav](https://arxiv.org/abs/2207.04429), where the authors saw a 2x improvement in detection with the large CLIP-ViT v/s ViLD.

- It's a bit misleading to call the task "robotic sorting", which implies a more embodied task and is already a fairly common phrase for the physical task of robotics sorting objects: e.g. [this](https://retinagan.github.io) and [this](https://arxiv.org/abs/1710.01330). A better scoping would help convey the contributions better, e.g. "detection", "identification" or "categorization".


- It would help to have a description or examples of the failure modes of the two-step approach, perhaps even a systematic analysis. Which of the two components is more vulnerable?

---

### Decision · Program_Chairs · 2022-11-15

Accept (Poster)